# Unraveling the spatial diversity of Indian precipitation teleconnections via nonlinear multi-scale approach

Jürgen Kurths[1,2,3], Ankit Agarwal[1,2,4], Roopam Shukla[1], Norbert Marwan[1], Maheswaran Rathinasamy[5], Levke Caesar[1,6], Raghvan Krishnan[7], and Bruno Merz[2,4]

[1]Potsdam Institute for Climate Impact Research (PIK), Member of the Leibniz Association, Telegrafenberg, Potsdam,
Germany
[2]Institute of Earth and Environmental Science, University of Potsdam, Potsdam, Germany
[3]Institute of Physics, Humboldt Universität zu Berlin, Germany
[4]GFZ German Research Centre for Geosciences, Section 5.4: Hydrology, Telegrafenberg, Potsdam, Germany
[5]Dept. of Civil Engineering, MVGR College of Engineering, Vizianagaram, India
[6]Institute of Physics and Astronomy, University of Potsdam, Potsdam, Germany
[7]Indian Institute of Tropical Meteorology, Pune, India

[*]Corresponding author at: agarwal@pik-potsdam.de/ aagarwal@uni-potsdam.de

**Abstract**

A better understanding of precipitation dynamics in the Indian subcontinent is required since India's society depends heavily on reliable monsoon forecasts. We introduce a nonlinear, multiscale approach, based on wavelets and event synchronization, for unraveling teleconnection influences on precipitation. We consider those climate patterns with highest relevance for Indian precipitation. Our results suggest significant influences which are not well captured by only the wavelet coherence analysis, the state-of-the-art method in understanding linkages at multiple time scales. We find substantial variation across India and across time scales. In particular, El Niño/Southern Oscillation (ENSO) and the Indian Ocean Dipole (IOD) mainly influence precipitation in the southeast at interannual and decadal scales, respectively, whereas the North Atlantic Oscillation (NAO) has a strong connection to precipitation particularly in the northern regions. The effect of PDO stretches across the whole country, whereas AMO influences precipitation particularly in the central arid and semi-arid regions. The proposed method provides a powerful approach for capturing the dynamics of precipitation and, hence, helps improving precipitation forecasting.

## 1. Introduction

Understanding the spatial patterns, frequency and intensity of precipitation in the Indian subcontinent is an active area of research due to its essential impact on life and property (Pai et al., 2015). The Indian monsoon is the pulse and lifeline of over one billion people, and the socio-economic development in this part of the world heavily depends on reliable predictions of the monsoon (Goswami and Krishnan, 2013; Shukla et al., 2018).

Numerous studies have emphasized the importance of understanding the influence of large-scale climatic patterns on precipitation for improving forecast accuracy (Feng et al., 2016), therefore many studies have analyzed the relationship between precipitation and climatic patterns for India. This research has shown that the relevant patterns are the El Niño/Southern Oscillation (ENSO) (Kumar et al., 2006; Mokhov et al., 2012), the Indian Ocean Dipole (IOD) (Behera et al., 1999; Krishnan and Swapna, 2009), the North Atlantic Oscillation (NAO) (Bharath and Srinivas, 2015; Feliks et al., 2013), the Pacific Decadal Oscillation (PDO) (Dong, 2016; Krishnan and Sugi, 2003), and the Atlantic Multidecadal Oscillation (AMO) (Goswami et al., 2006; Krishnamurthy and Krishnamurthy, 2016).

Over the years, linkages between climatic patterns and precipitation have been investigated by a range of statistical methods, such as correlation (Abid et al., 2018), principal component analysis (Luterbacher et al., 2006), empirical orthogonal functions (Hannachi et al., 2007), regression and canonical analysis (Xoplaki et al., 2004). However, all these methods are limited in capturing the scale-specific feedbacks and interactions between the long-range climatic patterns and precipitation. Such information is very crucial since in climatic systems energy is stored and transported differently on different temporal scales, resulting from interactions of intertwined sub-components across a wide range of scales (Miralles et al., 2014; Peters et al., 2007). Multiscale interactions have therefore received extensive attention in the field of climate dynamics (Peters et al., 2007; Steinhaeuser et al., 2012) and have been proposed as a mechanism for triggering extreme events (Agarwal et al., 2018b; Okin et al., 2009; Paluš, 2014; Peters et al., 2004) and abrupt transitions (Peters et al., 2007). That holds the promise of better understanding the system dynamics compared to analyzing processes at one time scale only.

In last decades, wavelet coherence has become the state-of-the-art method for studying the influence of climatic patterns on precipitation at different temporal scales. For example, Ouachani et al., (2013) investigated the multiscale linear relationship between Meditarrian region (Northen Africa) and large scale climatic patterns such as ENSO, NAO and PDO. The study reported the strong correlation between ENSO and precipitation series at lag of 2 years. The study further reported that the influence of ENSO on precipitation was stronger compared to other climatic modes considered in this particular study. Coherently, Tan et al., (2016) analyzed the relations between Canadian precipitation and different global climate indices. Similar studies using wavelet coherence also reported in other parts of the world (Agarwal et al., 2016; Araghi et al., 2017; Hu and Si, 2016; Tan et al., 2016). Though all such studies based on wavelet coherence were illuminating and contributed significantly in our existing knowledge of climate. However, there remains a large scope of advancement, that in particular in capturing the nonlinear scale-specific interactions between climate patterns and Indian precipitation.

To capture such nonlinear scale specific interactions, recently event synchronization (ES) has emerged as a powerful similarity measure (Agarwal et al., 2019a; Mitra et al., 2017; Ozturk et al., 2018; Quiroga et al., 2002) because ES automatically classifies pairs of events arising at two locations as temporally close (and, thus, possibly statistically – or even dynamically – interrelated) without the necessity of selecting an additional parameter in terms of a fixed tolerable delay between these events. Also, ES is a robust measure to study interrelationship between series of non-Gaussian data or data with heavy tails (Agarwal, 2019). These intrinsic features of ES are advantageous in climate in

general and to quantify interactions between climatic patterns and precipitation in particular since the time delay between such patterns (for e.g. ENSO) and their effect on precipitation is tedious to quantify beforehand.

We, therefore, decided to use ES to quantify the (possibly nonlinear) linkages between large-scale climatic patterns and precipitation across India. More specifically, we analyze the linkages between the 95-percentile extreme events, extracted from gridded Indian precipitation data at monthly resolution, and the climate patterns ENSO, IOD, NAO, PDO, and AMO which have been shown to be of significant relevance for precipitation in India. We combine ES with the wavelet transform, as proposed recently (Agarwal et al., 2017). This combination, termed MSES (Multiscale Event Synchronization), allows studying nonlinear connections between times series at different temporal scales. To consider the spatial variation across India, we sub-divide India into homogeneous regions that share rather similar precipitation characteristics and identify a representative grid cell for each region. The homogenous regions and the representative grid cells are obtained using the concept of complex networks approach (Agarwal et al., 2018a) and the resultant network is referred as climate network (Agarwal et al., 2019a; Boers et al., 2019; Ekhtiari et al., 2019; Tsonis et al., 2006).

The novelty of this study is the integration of (1) the nonlinear method for quantifying the linkages between large-scale climate patterns and climate network (precipitation) in India at (2) multiple time scales, considering (3) the spatial variation of these linkages. To our knowledge, this combination (nonlinear – multiple time scales – spatial variation) has not yet been implemented, neither for India nor for any other region. We argue that it allows unraveling the spatio-temporal diversity of Indian precipitation teleconnections, offering a compelling perspective for capturing the dynamics of precipitation and improving precipitation forecasting.

## 2. Study area and data

### 2.1. Study area

Our study area is the Indian subcontinent which shows a significant variation in climate characteristics. India extends over an area of 3,287,263 km$^2$. Its climate regimes are classified as arid (northwestern India), semi-arid (northern lowlands and central peninsular India), humid (coastal lowlands, southwestern and northeastern highlands) and alpine (Himalayan mountains in the north). The spatio-temporal variation of precipitation, as well as temperature, is significant over the country (Bharath and Srinivas, 2015). The entire country receives 80% of its total precipitation during the southwest monsoon, from June to September (Bharath and Srinivas, 2015). During the northeast monsoon (October to December), the precipitation is considerable but confined to the southeastern part of the country.

### 2.2. Gridded precipitation data

We use the high-resolution (0.25° × 0.25°) monthly gridded precipitation data set for the period 1951–2013, developed by the Indian Meteorological Department (IMD) for the spatial domain of 66.5°E to 100°E and 6.5°N to 38.5°N, covering the mainland region of India (Pai et al., 2014). The gridded data has been generated from the observations of 6995 gauging stations across India (Pai et al., 2014). The dataset captures well the spatial

distribution of precipitation over the country. For our study, out of 17415 grid cells, 4631 cells lying inside the boundaries of India were identified.

## 2.3. Time series of global and regional climate indices

For understanding the linkages between climate patterns and precipitation, we use time series of global and regional
climate indices for the same period, i.e., 1951–2013. We have selected those indices for which earlier studies have shown a relation to Indian precipitation. The selected climate indices and the respective studies are: ENSO (Mokhov et al., 2012), IOD (Ashok et al., 2001), NAO (Bharath and Srinivas, 2015; Feliks et al., 2013), PDO (Dong, 2016; Krishnan and Sugi, 2003) and, AMO (Goswami et al., 2006; Krishnamurthy and Krishnamurthy, 2016). For detailed information on these climate indices and the data sources, we refer to https://www.esrl.noaa.gov/.

## 3.   Methodology

To investigate the nonlinear, multiscale linkages between climate patterns and precipitation we propose an analysis based on combining network reconstruction, community detection, wavelet transformation, and event synchronization (Fig.1). First, we construct a precipitation network of the precipitation dataset using event synchronization. We further pool grid cells with similar precipitation characteristics into homogenous regions and
identify a representative grid cell for each region as proposed by Agarwal et al. (2018). The linkages between the precipitation time series of the representative cells and the teleconnection indices are analyzed by the Multiscale Event Synchronization (MSES) method developed by Agarwal et al. (2017). Finally, the proposed methodology is compared to the state-of-the-art wavelet coherence analysis (WCA).

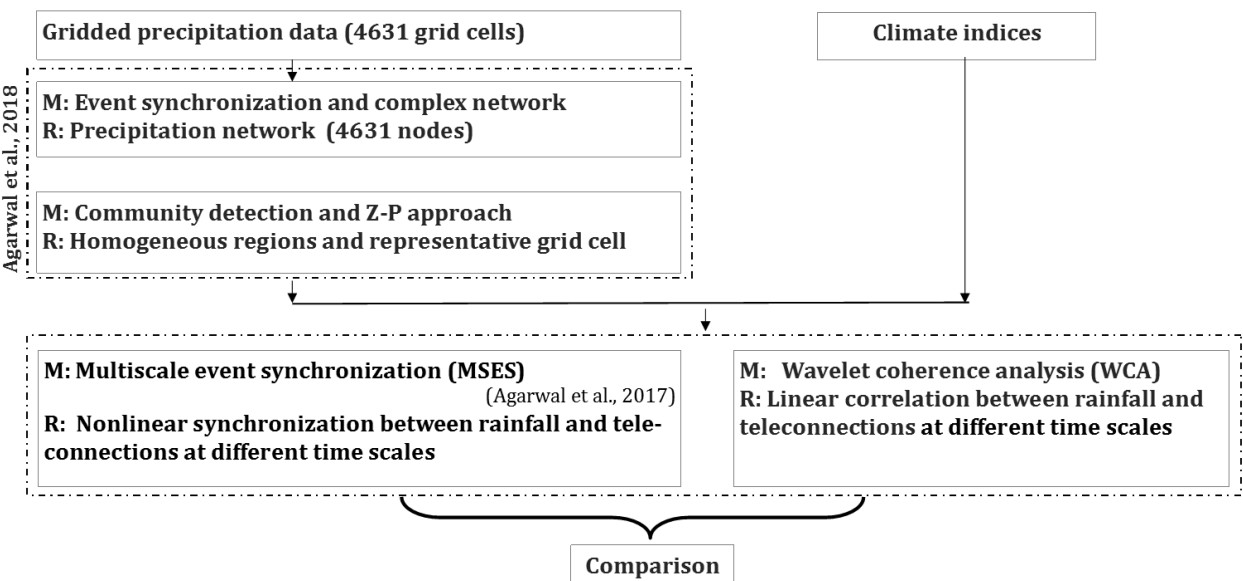

**Fig. 1 Schematic of the methodology to investigate the linkages between climate patterns and precipitation. (M stands for method and R for result.)**

### 3.1. Event synchronization and network construction

Event synchronization (ES) measures the nonlinear synchronization of point processes (Quiroga et al., 2002). Each grid cell of the precipitation data set serves as a network node, while the precipitation estimates at each cell provide the time series for that node. Following Agarwal et al. (2018), we define heavy precipitation events at each node as event with precipitation larger than the 95th percentile at that grid cell. The 95[th] percentile threshold for event selection is globally accepted which tradeoff between sufficient number of events and high threshold value. Then ES is used to calculate the strength of synchronization ($Q$) between all possible pairs of grid cells. ES has advantages over other time-delayed correlation techniques (e.g., Pearson lag correlation), as it uses a dynamic (not fixed) time delay (Agarwal et al., 2017). The latter refers to a time delay that is adjusted according to the two time series being compared, which allows its application to different situations. Another advantage of ES is that it can be applied to non-Gaussian data. Having its roots in neuroscience, ES only considers events beyond a threshold and ignores the absolute magnitude of events, which could be a challenge to incorporate in future work.

Here, we define an event when a value in the signal $x(t)$ (or $y(t)$) exceeds a threshold (selected by a $\alpha$ percentile). Events in $x(t)$ and $y(t)$ occurring at time $t_l^x$ and $t_m^y$ where $l = 1,2,3,4 \dots S_x$, $m = 1,2,3,4 \dots S_y$, are considered to be synchronized when they occur within a time lag $\pm\tau_{lm}^{xy}$ which is defined as (Agarwal et al., 2017b)

$$\tau_{lm}^{xy} = min\left\{t_{l+1}^x - t_l^x, t_l^x - t_{l-1}^x, t_{m+1}^y - t_m^y, t_m^y - t_{m-1}^y\right\}/2 \tag{1}$$

$S_x$ and $S_y$ are the total number of events (greater than threshold $\alpha$) that occurred in the signal $x(t)$ and $y(t)$, respectively. This definition of the time lag helps to separate independent events. Then we count the number of times an event occurs in the signal $x(t)$ after the maximum time lag $\tau_{lm}^{xy}$ of an event that appears in the signal $y(t)$ and vice versa, resulting in the quantities C(x|y) and C(y|x):

$$C(x|y) = \sum_{l=1}^{S_x}\sum_{m=1}^{S_y} J_{xy} \quad and \quad C(y|x) = \sum_{l=1}^{S_x}\sum_{m=1}^{S_y} J_{yx} \tag{2}$$

with

$$J_{xy} = \begin{cases} 1 & if \ \ 0 < t_l^x - t_m^y < \tau_{lm}^{xy} \\ \frac{1}{2} & if \ \ \ \ \ \ \ \ \ \ \ \ \ t_l^x = t_m^y \\ 0 & else, \end{cases} \tag{3}$$

From these quantities, we define a measure of the strength of event synchronization ($Q_{xy}$) between $x(t)$ and $y(t)$ by

$$Q_{xy} = \frac{C(x|y) + C(y|x)}{\sqrt{(S_x - 2)(S_y - 2)}}. \tag{4}$$

$Q_{xy}$ is normalized to $0 \le Q_{xy} \le 1$. $Q_{xy} = 1$ refers to perfect synchronization between the signals $x(t)$ and $y(t)$. Event synchronization (ES) has been specifically designed to identify nonlinear associations among event time series with varying lags between them.

A link between two grid cells is set up if their heavy precipitation occurrences are strongly synchronized, which we define as having a $Q$ value greater than predefined threshold ($\theta_{x,y}^{Q}$). A number of criteria have been proposed to generate an adjacency matrix from a similarity matrix, such as a fixed amount of link density (Agarwal, 2019) or fixed thresholds (Donges et al., 2009). Here, we consider a 5% link density since it is a well-accepted criteria in general for the network construction. Also, 95[th] percentile is a good trade-off between sufficient number of connections and capturing high synchronized connections.

We repeat the procedure for all possible pairs of nodes to construct a precipitation network with the adjacency matrix

$$A_{x,y} = \begin{cases} 1, & if \ Q_{x,y} \geq \theta_{x,y}^{Q} \\ 0, & else. \end{cases} \tag{5}$$

Here, $\theta_{x,y}^{Q} = 95^{th}$ percentile is a chosen threshold, and $A_{x,y} = 1$ denotes a link between the $x^{th}$ and $y^{th}$ nodes and 0 denotes otherwise. The adjacency matrix represents the connections in the rainfall network. In this study, we use an undirected network, meaning we do not consider which of the two synchronized events happened first, in order to avoid the possibility of misleading directionalities of event occurrences between nodes that are topographically close to one another.

### 3.2. Community detection and Z-P approach

The linkages between climate indices and precipitation are evaluated on a regional scale. India is subdivided into homogeneous regions with similar characteristics of heavy precipitation events using the concept of complex networks (Agarwal et al., 2018). Several studies such as (Agarwal et al., 2019b; Halverson and Fleming, 2015; Lancichinetti and Fortunato, 2009; Newman, 2006; Sivakumar et al., 2015; Tsonis et al., 2011) have reported superior performance of complex networks in identifying homogeneous regions compared to more traditional methods, such as the hierarchical clustering algorithm or the information-theoretic algorithm (Harenberg et al., 2014).

There exist several community detection approaches aiming at stratifying the nodes into communities in an optimal way (see Fortunato, (2010) for an extensive review). The question which community detection algorithm should be used is difficult to answer. However, it has been found that the choice of the community detection algorithm has a small impact on the resultant communities in geophysical data science studies (Halverson and Fleming, 2015). In this study, we use the Louvain method which maximizes the modularity to find the optimal community structure in the network. The optimal community structure is a subdivision of the network into non-overlapping groups of nodes, which maximizes the number of within-group edges and minimizes the number of between-group edges (Blondel et al., 2008; Rubinov and Sporns, 2011).

Modularity is defined, besides a multiplicative constant, as the number of edges falling within groups minus the expected number in an equivalent network with edges placed at random. Positive modularity values suggest the presence of communities. Thus, one can search for community structures by looking for the network divisions that have positive, and preferably large, modularity values (Newman, 2004). Modularity (*M*) is calculated as:

$$M = \frac{1}{2m} \sum_{x,y} \left| A_{xy} - \frac{k_x k_y}{2m} \right| \delta(C_x C_y) \tag{6}$$

where $A_{xy}$ represents the number of edges between $x$ and $y$, $k_x = \sum_y A_{xy}$ is the sum of the number of the edges (degree) attached to vertex $x$, $C_x$ is the community to which vertex $x$ is assigned, the $\delta -$ function $\delta(u,v)$ is 1 if $u = v$ and 0 otherwise, and $m = {}^1/_2 \sum_{xy} A_{xy}$.

Equation (6) is solved using the two-step iterative algorithm proposed by Blondel et al. (2008), also known as the Louvain method. The first step consists in optimizing the modularity by permitting only a local modification of communities; in the second step, the communities identified are pooled to assemble a new network of communities. High modularity networks are densely linked within communities but sparsely linked between communities. The algorithm stops when the highest modularity is achieved.

Further, for each community, we identify a representative grid cell using the Z-P space approach, where $Z$ is within module-degree or Z-score and $P$ is the participation coefficient (Agarwal et al., 2018). The within-module degree ($Z_x$ or Z-score) is a within-community version of degree centrality (total number of link of any node) and shows how well a node is connected to other nodes in the same community. It is estimated as (Guimer and Amaral, 2005)

$$Z_x = \frac{K_x - \overline{K_{s_x}}}{\sigma_{k_{s_x}}} \tag{7}$$

where $K_x$ is the total number of links (degree) of node $x$ in the community $s_x$, $\overline{K_{s_x}}$ is the average degree of all nodes in the community $s_x$, and $\sigma_{k_{s_x}}$ is the standard deviation of $K$ in $s_x$. Since two nodes having the same Z-score may play different roles within the community, this measure is often combined with the participation coefficient $P_x$.

The participation coefficient ($P_x$) compares the number of links of node $x$ to nodes in all communities with the number of links within its own community. We define the $P_x$ of node $x$ as (Guimer and Amaral, 2005)

$$P_x = 1 - \sum_{s_y=1}^{N_M} \left( \frac{k_{xs_y}}{k_x} \right)^2 \tag{8}$$

where $k_{xs_y}$ is the number of links of node $x$ to nodes in community $s_y$, and $k_x$ is the total number of links (degree) of node $x$. $N_M$ represent the number of communities in the network. The participation coefficient of a node is therefore close to one if its links are uniformly distributed among all the communities, and zero if its entire links are within its own community because in later case $K_{xs_y} = K_x$ hence $P_x = 0$.

The cell with the highest number of intracommunity links is considered as representative (Halverson and Fleming, 2015), based on the argument that this cell shows the strongest synchronization within the community. We expect its climatological properties, such as the linkage to large-scale climate patterns, to have the highest similarity to the properties of the other cells in the community. We could also use a composite, e.g., by normalizing the grid cell time series and defining the time series of the mean of the normalized series as representative. However, this definition would reduce the variability and could mask existing connections to climatic patterns.

### 3.3. Multiscale Event Synchronization

In this study, we use the multiscale event synchronization (MSES) measure (Agarwal et al., 2017), that combines the wavelet transform and event synchronization, to quantify the relationship between precipitation and climate indices. The multiscale event synchronization measure is based on a combination of wavelet transform and event synchronization (section 3.1). The following subsections discuss briefly the wavelet transform and finally the methodology for MSES.

*Wavelet Analysis*

Synthetically the temporal data series of any continuous geophysical variable is the superposition of variations occurring at different scales. Different physical processes drive these patterns, and a partitioning of the variability at different scales can help to isolate and characterize underlain processes (Sturtevant et al., 2015). Wavelets have been successfully used to characterize the time scale of interactions across fluxes and physical drivers (Katul et al., 2001, Ding et al., 2013).

The wavelet transform of a signal decomposes it into a set of components with predefined central frequencies and spectral bandwidths. Here we use the Maximal Overlap discrete wavelet transform (MODWT) (Percival and Walden, 2000) because the orthogonal discrete wavelet transform (DWT) results in a pyramid of wavelet coefficients which does not contain the time synchronization of the events. Further, our experience with DWT suggests that the latter approach suffers from 'shift sensitivity' also known as 'shift variance' and is undesirable because it implies that DWT coefficients fail to distinguish between input-signal shifts (Maheswaran and Khosa 2012). Even though the MODWT has large redundancy, it is shift invariant, and this property renders the MODWT more suited for time series analysis.

MODWT decomposes the time series into different time scales or frequency components. The wavelet decomposition is realized using the two basis functions known as father wavelets and mother wavelet. Any function $f(t)$ can be expressed in these basis functions and their scaled and translated versions as given in Eq.(9)

$$f(t) = \sum_k s_{J,k}\, \varphi_{J,k}(t) + \sum_k d_{J,k}\, \psi_{J,k}(t) + \sum_k d_{J-1,k}\, \psi_{J-1,k}(t) \dots \dots + \sum_k d_{1,k}\, \psi_{1,k}(t) \tag{9}$$

where $J$ is the number of multiresolution components (scales), and $k$ is in the rangeof 1 to the number of coefficient in the specified component. The coefficients $s_{J,k}$ are the approximation coefficients and $d_{J,k}, \dots, d_{1,k}$ are the wavelet transform coefficients, while the functions $\varphi_{J,k}(t)$ and $\{\psi_{j,k}(t))| j = 1, \dots, J-1, J\}$ are the approximating wavelet function and detailed wavelet functions respectively.

These basis functions are defined in terms of father and mother wavelets as follows:

$$\varphi_{j,k}(t) = 2^{-j/2}\varphi(2^{-j}t - k) \tag{10}$$

$$\psi_{j,k}(t) = 2^{-j/2}\psi(2^{-j}t - k) \tag{11}$$

Further,

$$s_{J,k} \approx \int \varphi_{J,k}(t)f(t)dt \ , \tag{12}$$

$$d_{j,k} \approx \int \psi_{J,k}(t)f(t)dt \ , \ j = 1, \dots J-1, J \tag{13}$$

where the scaling coefficients $s_{J,k}$ capture the smooth trend of the time series at the coarse scale $2^J$, which are also called smooth coefficients; and the wavelet coefficients $d_{j,k}$, also known as detail coefficients can detect deviations from the coarsest scale to the finest scale.

The original series $f(t)$ can be reconstructed by the summing the detailed components and the smooth components.

$$f(t) = S_{J,k} + D_{J,k} + D_{J-1,k} + \cdots \dots \dots D_{1,k} \tag{14}$$

where

$$S_{J,k} = \sum_k s_{J,k}\varphi_{J,k}(t) \ , D_{J,k} = \sum_k d_{J,k}\psi_{J,k}(t) \ , \dots \dots , D_{1,k} = \sum_k d_{1,k}\psi_{1,k}(t) \tag{15}$$

Eq (14) defines a multiresolution analysis (MRA) of $f(t)$; i.e., we express the series f(t) as the sum of a constant vector $S_J$ and J other vectors $D_j$, $j = 1, \dots, J$, each of which contain a timeseries related to variations in f(t)at a certain scale. We refer to $D_j$ as the $j^{th}$ level wavelet detail. Figure 2 shows the MODWT decomposition of a sample signal up to 7 scales resulting in 7 detailed components (D1- D7) and one approximate Component (S7).

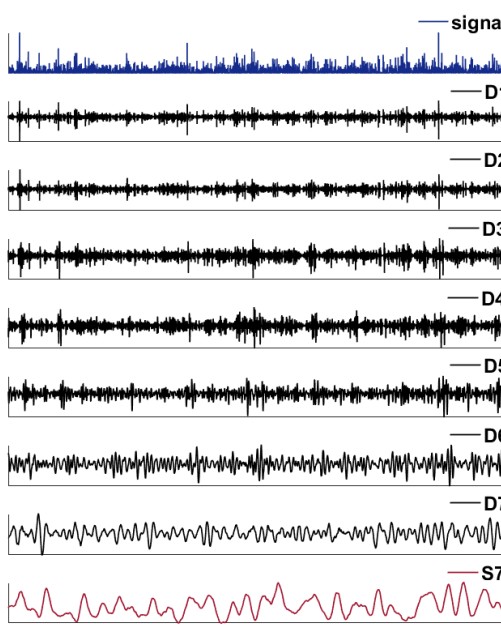

**Figure 2 Scheme of multi-scale decomposition of signals using maximum overlap discrete wavelet transformation (MODWT). The relationship between signal $Y_t$ (blue), detailed component $D_j$ (black), and approximate component $S_j$ (red), is shown.**

Let $Y_t$ represents a time series history of a geophysical process. In order to partition the variability of the process at different scales $j = 1 \dots J$, the signal $Y_t$ is transformed into the wavelet space which provides the required
information at different scales. This is obtained by convolving $Y_t$ with a set of low pass ($g$) and high pass ($h$) filters.

For instance, at each scale $j$, the MODWT applies a high pass wavelet filter $h_{j,l}$ and a lower pass filter $g_{j,l}$ of length ($l$) to the time series $Y$ to respectively yield the wavelet coefficients $W_{j,t}$ and $V_{j,t}$ for every point $t$ in the time series (Percival and Walden 2000).

$$\left. \begin{array}{l} W_{j,t} = \sum_{l=0}^{L_j-1} h_{j,l} Y_{t-l \bmod 2N} \\ V_{j,t} = \sum_{l=0}^{L_j-1} g_{j,l} Y_{t-l \bmod 2N} \end{array} \right\} \tag{16}$$

The $W_{j,t}$ wavelet coefficients distinguish fluctuations in the time series of scale $2^{j-1}$, while the $V_{j,t}$ coefficients provide information about the variations at scale $2^j$ and higher. Let the maximum level of decomposition be $j = J$. This would result in a total $'J + 1'$ series of wavelet coefficients with $W_{j,t}, j = 1,2,3 \dots J$, and one series of $V_{J,t}$.

Let us now define $D_j$ which represents the time domain reconstruction of $W_j$. It represents the portion of $Y$ attributable to scale $j$. Let $S_J$ represent the time domain reconstruction of $V_J$. For the maximum level of decomposition, $V_J$ has all of its elements equal to the sample mean of $Y$.

Therefore, we can write

$$Y = \sum_{j=1}^{J} D_j + S_J \tag{17}$$

**Stepwise procedure to estimate MSES**

The MSES values between precipitation and climate indices are estimated in the following manner:

a. The climate indices and precipitation values at monthly resolution are decomposed into its various scale specific components as proxies of the corresponding signal using the maximum overlap discrete wavelet transformation (MODWT). These components represent the features of the signal at different time scales. We limit the analysis to scale 7, i.e., 16 years, due to the distortion created by the boundary effects of the wavelet decomposition (Percival, 2008).

b. After fixing a 95% threshold for each of the decomposed components of precipitation and climate indices, the event synchronization values are estimated. The 95% threshold values are estimated for each scale component separately, ensuring a reliable estimation of the synchronization between the events.

c. The estimated ES values are considered significant if they are higher than the ones obtained from a significance test (Agarwal et al., 2017).

d. These steps (a-c) are repeated for all combinations of climate indices and precipitation for the different regions.

### 3.4. Significance test for similarity measure

To evaluate the statistical significance of the ES values, a surrogate test is used as proposed by Agarwal et al., 2017. We randomly reshuffle each time series 100 times (arbitrary number) but keeping the distribution same. The reshuffling will ensure that any potential synchronization between the even series will be destroyed and that they will be equivalent to independent random series. Then, for each pair of time series (rainfall and climate time series), we calculate the MSES values for the different scales. At each scale, the empirical test distribution of the 100 MSES values for the reshuffled time series is compared to the MSES values of the original time series. Using a 1% significance level, we assume that synchronization cannot be explained by chance if the MSES value at a certain scale of the original time series is larger than the $99^{th}$ percentile of the test distribution.

### 3.5. Testing of MSES on a synthetic dataset

In a former study, we have tested the MSES measure with different synthetic time series and have shown the efficacy of the method (Agarwal et al., 2017). In this paper, we further test the method with synthetic data sets and compare the results with those of the traditional methods such as correlation analysis and wavelet coherence.

Consider two time series $X$ and $Y$ (of length 1000) as defined by

$$X(t) = sin(0.5t) + e(t) + a * H(t) \tag{18}$$
$$Y(t) = sin(0.1t) + e(t) + a * H(t-3) \tag{19}$$

where $e(t)$ and $H(t)$ denotes a white noise process $\sim N(0,1)$ and a random binary series with values 1 or 0. Respectively $H(t)$ represents the aperiodic extreme events in the given time series.

Figure 3 shows the plot of $X$ and $Y$ with respect to time. It can be observed that $X$ and $Y$ are time series with two distinct frequencies but have a lagged relationship induced by $H(t)$. In this example, we have considered $a$=4. The zero lag correlation coefficient between $X$ and $Y$ can be estimated as –0.02 and the Pearson lag correlation is found to be 0.3, both showing no significant correlation. However, as expected the MSES given by Agarwal et al., 2017; was estimated to be 0.9375 at scale 1 revealing the underlying synchronization between the two series in this scale only.

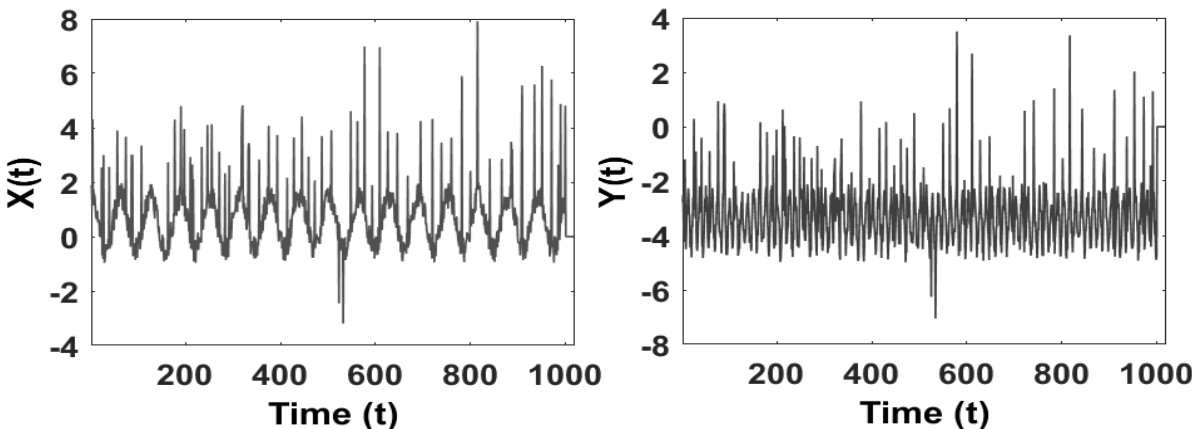

**Figure 3 Test signal $X(t)$ (left) and $Y(t)$ (right) with two distinct frequencies having lagged relationship induced by $H(t)$ used to explain the methodology.**

**3.6. Wavelet coherence analysis (WCA)**

We benchmark the MSES results against the wavelet coherence analysis (WCA) analysis because wavelet coherence is the state-of-the-art method in evaluating linkages between hydroclimatological variables at multiple time scales (Peters et al., 2004; Tan et al., 2016). We use the Grinsted Toolbox (Grinsted et al., 2004) for calculating the WCA between precipitation of the representative grid cells and the climatic indices. The wavelet coherence between time series {$X_t$} and {$Y_t$} was defined by (Torrence and Compo, 1998) as

$$R^2(j,t) = \frac{\left| \zeta \left( j^{-1} W_{xy}(j,t) \right) \right|^2}{\zeta \left( j^{-1} \left| W_x(j,t) \right|^2 \right) \zeta \left( j^{-1} \left| W_y(j,t) \right|^2 \right)} \tag{20}$$

Where $R^2(j,t)$ takes a value between 0 and 1; $\zeta$ is a smoothing operator and can be written as

$\zeta(W) = \zeta_{scale}(\zeta_{time}(W(j,t)))$. $W_{xy}$ represents the cross-wavelet coefficient between X and Y. $W_x$ (j,t) and $W_y$ (j,t) denote the wavelet coefficients obtained from wavelet transform of X and Y respectively at scale $j$ and time $t$. The global wavelet coherence at a certain scale $j$ is defined as the time-averaged value of the wavelet coefficients at the scale with the COI. It is estimated by

$$R^2(j) = \frac{1}{n_j} \sum_{t=t1}^{t2} R^2(j,t) \tag{21}$$

Where $n_j$ is the number of points with COI and $n_j = t2 - t1 + 1$.

Global wavelet coherence is a useful measure to examine the common characteristic periodicities between $x$ and $y$. Grinsted et al. showed the applicability of WC analysis of the association of precipitation with climate variables(Grinsted et al., 2004). A more detailed description of wavelet coherence analysis is can be found in (Grinsted et al., 2004).

It is important to note that WCA uses the complete, continuous time series for quantifying the linkages between precipitation and climate patterns, whereas MSES first derives extreme events at the different time scales, and then uses the synchronization between these events to identify the linkages.

## 4.  Results and Discussion

### 4.1  Homogeneous regions and representative grid cells

To reduce the number of pairs of precipitation and climate index time series for finding synchronization, we pool precipitation grid cells with similar heavy precipitation event characteristics into homogeneous regions. These regions and their main physical characteristics are given in Fig. 4. A more detailed discussion of these regions is provided in a previous study (Agarwal et al. 2018). For each community (C1 to C7), we identify a representative grid cell (black dots in Fig. 4) using the Z-P space approach. C1 and C2 (Fig.4) both are in South India but are differentiated by topological (elevation, land, coastline and climate regimes) features. C3 have moderate elevation, equatorial grasslands and semi-arid climate regimes. C4 covers almost all of the greenest and most mountainous regions of India (northeastern India). C5 is northwestern India covers dry and lowland areas. C6 in the western coastline is near to both coastlines and are low-lying areas with two different climate regimes (arid and humid). C7 is very high mountainous region with alpine climate regimes Next, we investigate the nonlinear linkages between the precipitation time series of the representative cells and the climate indices.

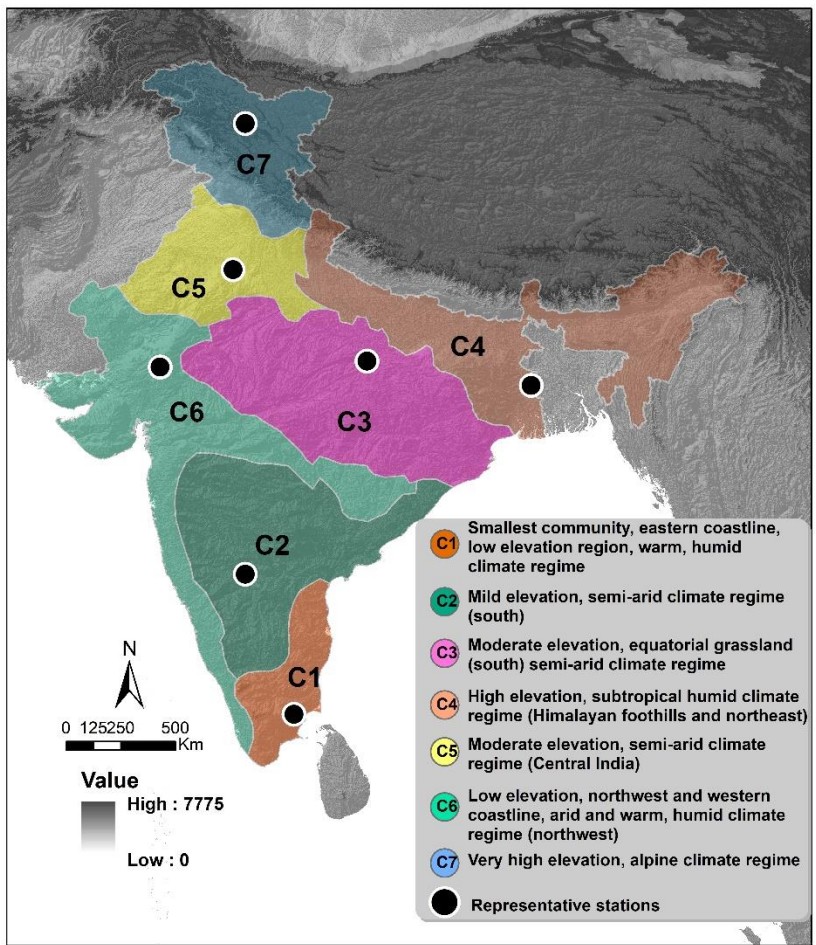

**Fig. 4 Spatial distribution/extent of the seven regions, or communities, with similar heavy precipitation event characteristics across India. Black dots indicate the representative grid cells for each of the community identified using the Z-P space approach. Terrain characteristics of the Indian subcontinent is shown using the SRTM DEM (in the background).**

**4.2 Linkages between precipitation and climatic patterns at multiple time scales**

Figs.5(a-e) and 6(a-e) show the MSES values and WCA values between precipitation and the climate indices, respectively. They are given for the five chosen climate indices and extreme precipitation in each of the representative grid cells of the seven homogeneous regions.

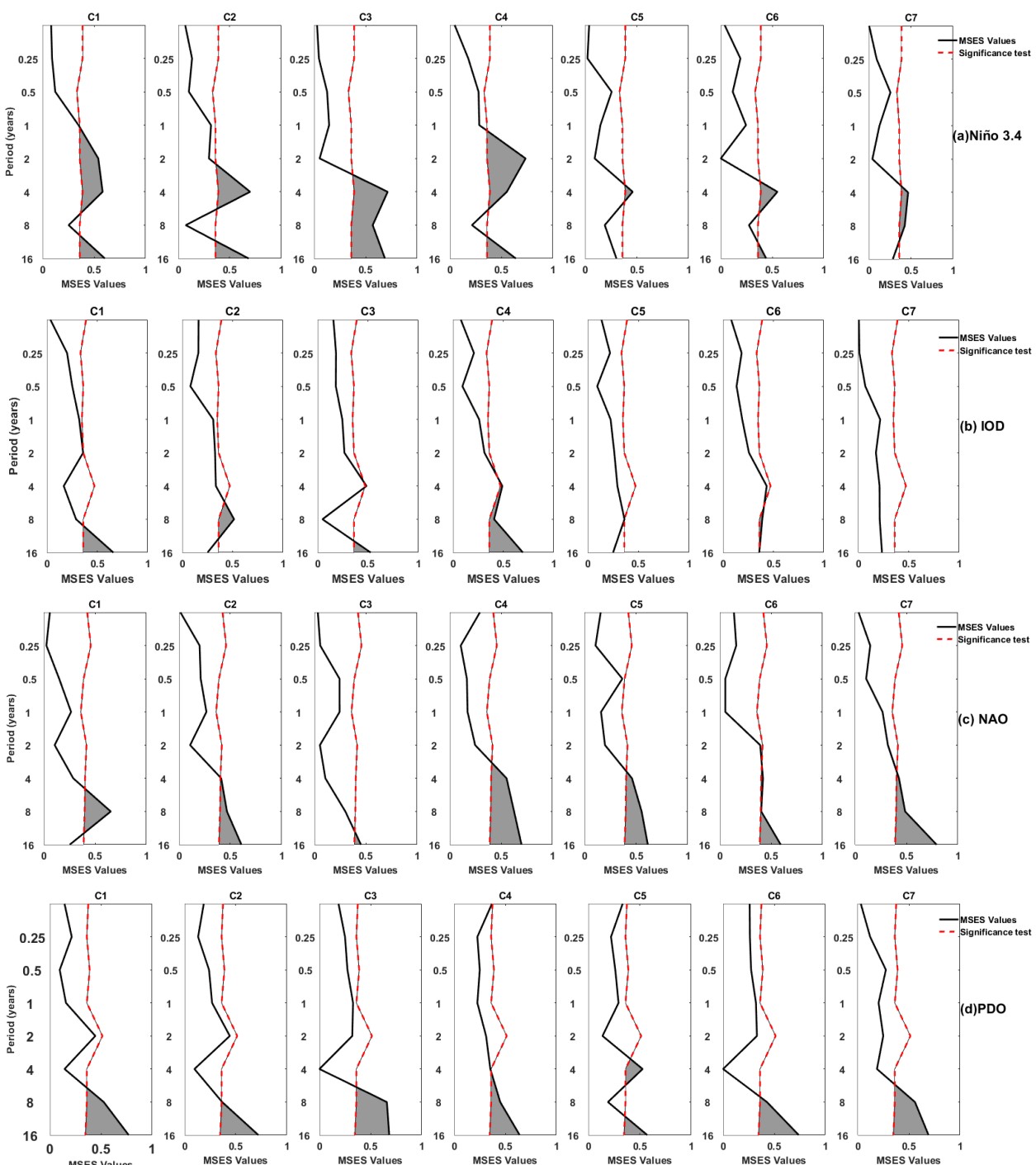

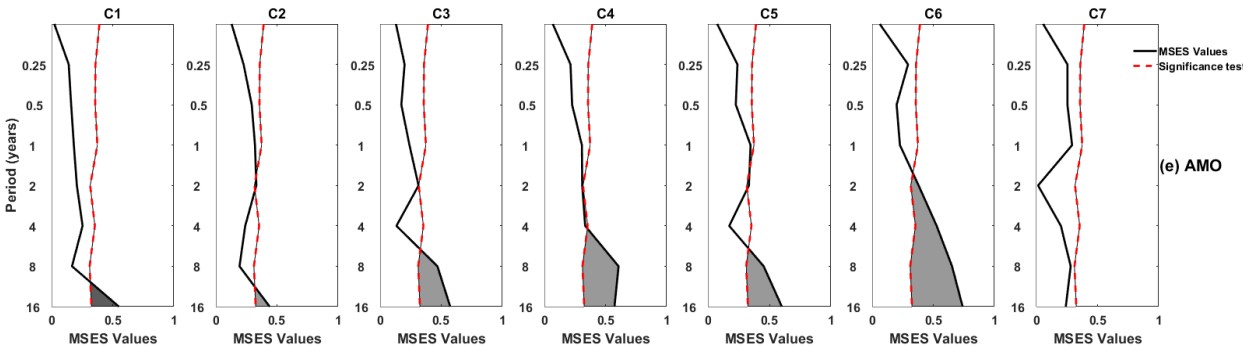

**Fig. 5 Multiscale event synchronization (MSES) between precipitation and climate indices. From top to bottom: Nino3.4, IOD, NAO, PDO, and AMO. From left to right: Community 1 to community 7. MSES values are shown as solid lines, and significant connections (at the 95% significance level) are marked in grey.**

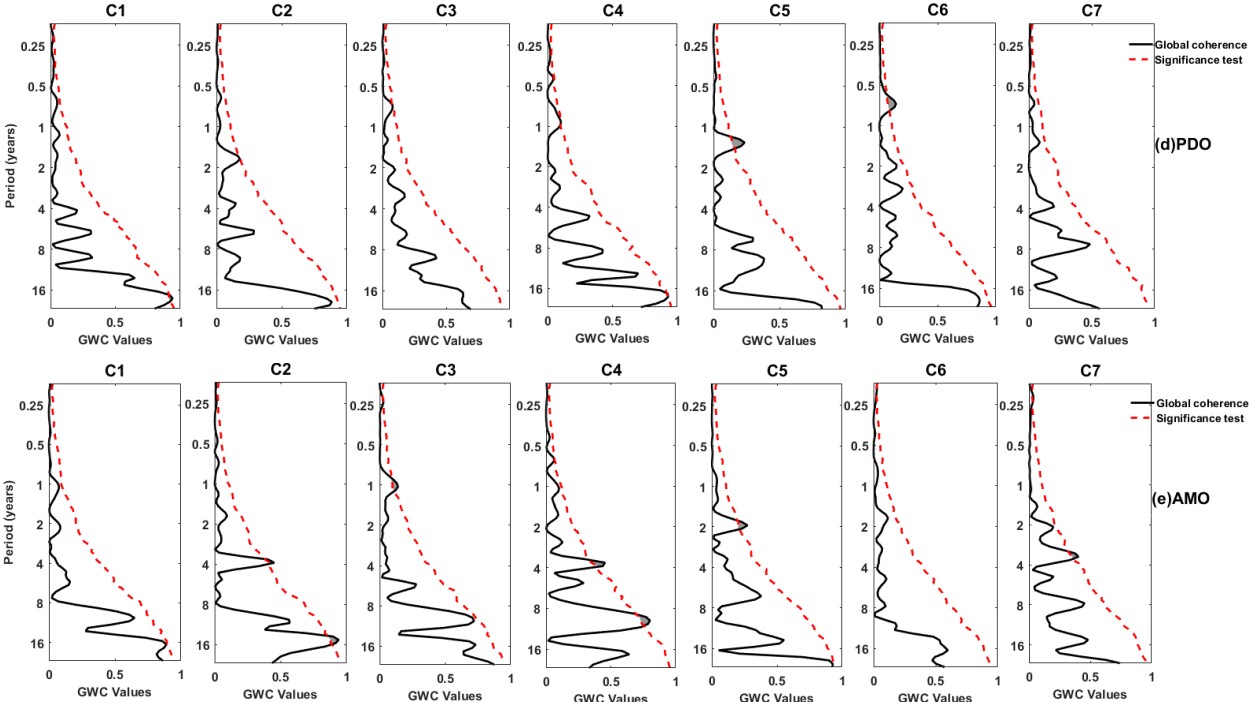

**Fig.6 Global Wavelet Coherence (GWC) between precipitation and climate indices. [Top to bottom]: Nino3.4, IOD, NAO, PDO, and AMO. From left to right: Community 1 to community 7. WCA values are shown as solid lines, and significant connections (at the 95% significance level) are marked in grey.**

Fig.5a shows a significant association between El Niño/Southern Oscillation (ENSO) and precipitation in all regions of India at the interannual scale. Its strength varies in space and with temporal scale. It is stronger for the southeastern peninsular (C1, C2, C3 and C4) and decreases notably in the northwestern Himalayan (C5, C6 and C7) regions. In the southeastern peninsula, the highest synchronization for the low (C1), mild (C2) and moderate (C3) elevation regions occurs at the 4-year scale, and at the 2-year scale for the high elevation (C4) region. For the southeast regions of India, we observe a significant synchronization at the decadal scale (8-16 years) which is counterintuitive given the interannual time scale of ENSO (D'Arrigo, 2005; McGregor et al., 2013). The analysis based on WCA (Fig. 6a) shows substantially less correlation between precipitation and ENSO in all regions.

Overall, the association between ENSO and precipitation at the interannual scale is coherent with the general understanding that extreme precipitation in India is associated with ENSO (Rajeevan and Pai, 2007). Additionally, our analysis reveals the important spatial variation of this linkage across India, which has not yet been reported before. We find stronger linkages for the regions close to the ocean (southeastern peninsular comprising C1 to C4) compared to the inland regions with higher elevation (northwestern India comprising C5, C7). The results mentioned above are in congruence with the findings by (Guhathakurta et al., 2017; Mishra et al., 2012). The spatial heterogeneity in the strength of the relationship between ENSO and precipitation may be a result of the tropical convection during the ENSO events (Bansod, 2011). Other studies have confirmed that there is a decrease in the strength of the relationship between precipitation and ENSO events with distance from the ocean. A similar pattern observed in Mexico where the Nino 3.4 teleconnection is weaker, if not opposite in sign, in northern versus southern

Mexico (Hu and Feng, 2002). This observation leads us to the understanding that the ENSO teleconnection is strong in regions of climatologically strong convection.

Interestingly, an association between ENSO and precipitation at the decadal scale has not been reported for India so far. This association might be a consequence of the interdependencies between ENSO and IOD at the decadal scale (Luo et al., 2010). Recently, Izumo et al., (2010), demonstrated that IOD events tend not only to co-occur with ENSO events but also to lead them through tropospheric biennial oscillation (Pillai and Mohankumar, 2010). MSES has the potential to capture such interdependencies when applied directly to such indices. However, this is beyond the scope of the study.

The synchronization and coherence between the Indian Ocean Dipole (IOD) and precipitation are given in Fig.5b and Fig.6b, respectively. The nonlinear dependence measure points to a significant synchronization at time scales of 8-16 years in the southeastern regions C1-C4. The rest of the country seems to be unaffected by IOD. The WCA analysis obtains a similar spatial pattern, however, the significant associations occur at shorter time scales (1-4 years, Fig. 6b). Interestingly, with both methods, we cannot find any coupling with the Himalayan region (C7).

The results obtained by MSES and WCA are in accordance with the general understanding that IOD plays a vital role in the Indian monsoon system in the southeast regions, i.e., in close proximity to the Indian Ocean, at interannual and decadal scales (Krishnan and Swapna, 2009). This result can be explained by the fact that one of the general conditions for Indian precipitation is the Tropical Easterly Jet and Tropical Westerly Jet (Rai and Dimri, 2017). In case of occurrence of IOD, the pressure dipole generated between the Tibetan plateau and the Madagascar Island either strengthens the southeastern Indian monsoon (positive IOD) or weakens it (negative IOD) (Jiang and Ting, 2017). However, the reason for the association at the decadal scale is not apparent and needs further investigation.

Unlike IOD, North Atlantic Oscillation (NAO) demonstrates significant synchronization with precipitation across the entire subcontinent (Fig.5c). The linkages to the northern regions C4, C5 and C7 are strong and significant at interannual and decadal scales, whereas the southern regions C1, C2, C3 and C6 show weaker linkages. Overall, the strength of synchronization between NAO and Indian extreme precipitation is higher at the decadal scale than at the interannual scales. The comparison of the results obtained by MSES (Fig.5c) and WCA (Fig.6c) reveals that the nonlinear method shows an increase in the association particularly in the northeastern Himalayan foothill region (C4). For some regions, MSES detects linkages which are not found by WCA. For example, in the Himalayan region (C7), MSES shows a significant association at time scales of 4-16 years, whereas WCA shows only a signal just at the significance level at the scale of 16 years. The overall MSES results are in congruence with other studies (Bhatla et al., 2016; Feliks et al., 2013; Goswami et al., 2006), but so far space and scale variation in the associations between NAO and Indian precipitation has not gained attention. The linkages between precipitation and NAO in the northern part of the country might be due to westerly influences from the Eurasian region which are, in turn, strongly affected by NAO. Another explanation (Goswami et al., 2006) suggests that the linkage of NAO and Indian precipitation at higher scales (decadal and beyond) in the northern part of India results from the interdependency of NAO and AMO.

In the case of Pacific Decadal Oscillation (PDO), we infer a robust decadal synchronization across the entire subcontinent (Fig.5d). The strength of synchronization varies in space and reaches values of around 0.7 for several regions. On the contrary, WCA (Fig.6d) does not reveal significant associations at the decadal scale except for the eastern coastline (C1) and Himalayan foothills (C4), where values at the boundary to significance are found.

The MSES results agree with Krishnan and Sugi (2003) who demonstrate a strong relationship between PDO and precipitation across the country. The interannual synchronization might be an indirect influence because of the interdependency of PDO and ENSO (Krishnan and Sugi, 2003; Rathinasamy et al., 2014).

The highest strength of synchronization between Atlantic Multidecadal Oscillation (AMO) and Indian precipitation is observed in the northwestern and central regions C3 to C6 (Fig. 5e). Weaker associations are detected in the south (C1, C2), whereas no significant synchronization is found for the Himalayan region (C7). The linkages are most prominent at the decadal scale; in some regions also significant synchronization at interannual scales is found. In contrast, WCA shows only weak linkages (Fig. 6e).

Our MSES results confirm the assertion given by Zhang and Delworth, (2005) who found an in-phase relationship between Indian precipitation and AMO. A study by Goswami et al., (2006) also unraveled a link between AMO and multidecadal variability of Indian precipitation. However, our study is the first to observe that the strength of the coupling between AMO and precipitation varies according to the different climate regions and is strongest at the decadal scale.

In summary, our findings re-confirm known physics-based associations, thus implicitly affirming the validity of our approach, but also provide new insights into Indian precipitation teleconnections. We find substantial spatial variation in the significant linkages across India and for different time scales (Fig.7). MSES reveals an appreciable increase in the association between climate patterns and precipitation in most regions when compared to WCA. In some regions, the synchronization values increase by 40-50%. The much higher skill of MSES in detecting associations suggests the presence of nonlinear and threshold relationships which can not be captured by WCA which is limited to linear processes.

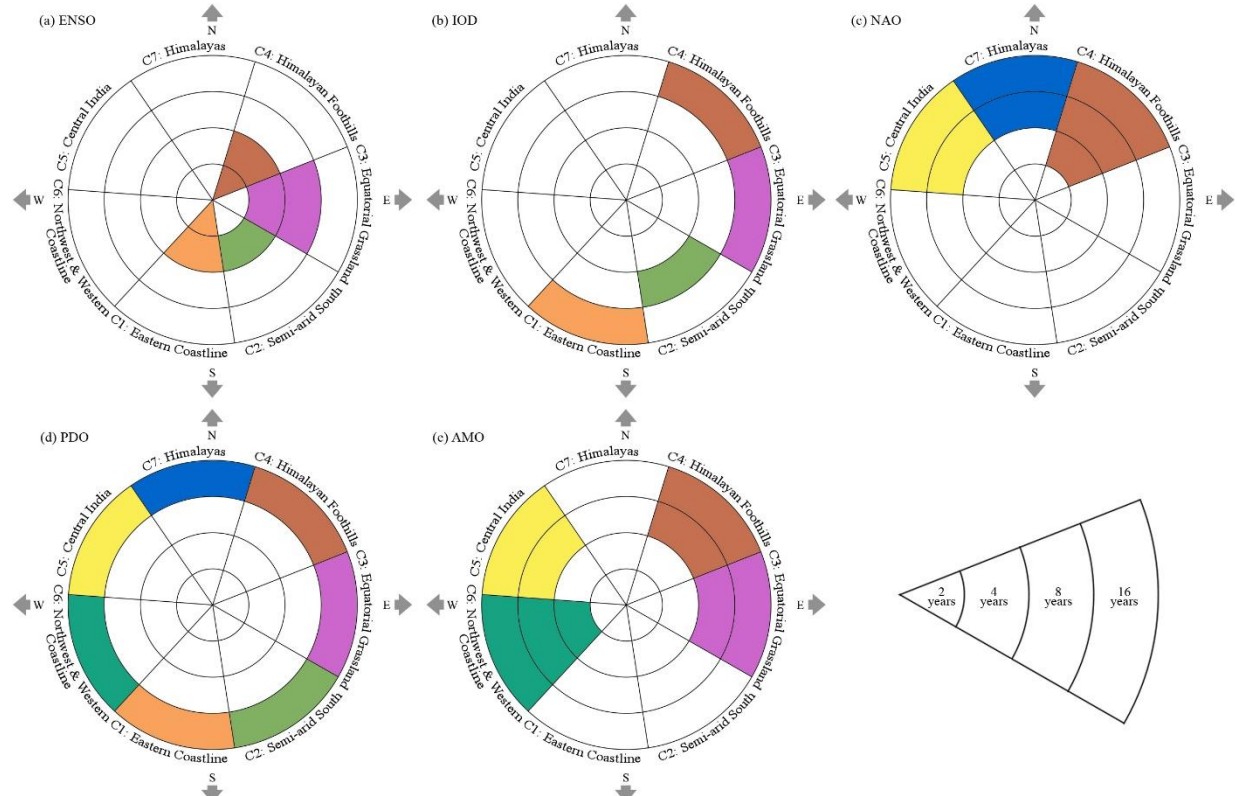

**Fig. 7 Schematic map of spatial diversity of Indian precipitation teleconnections at different time scales. (a) Nino3.4, (b) IOD, (c) NAO, (d PDO, and (e) AMO. Colors are consistent with the community shown in the Fig.4. Presence of color (irrespective of magnitude of synchronization) in community segment indicates significant synchronization between teleconnection and Indian precipitation. Every single segment of circle shows the temporal scale. Cardinal direction has been projected in the background of each circle.**

## 5. Conclusions

A novel nonlinear, multiscale approach (MSES) based on wavelets and event synchronization is used for unraveling teleconnection influences on Indian climate network at multiple time scales. The analysis considers those climate patterns with highest relevance for Indian precipitation. To understand the spatial heterogeneity, India is sub-divided into homogeneous regions using complex networks. The comparison with wavelet coherence analysis (WCA), the state-of-the-art method in understanding linkages at different time scales, shows a much higher skill for MSES in detecting linkages between climate indices and precipitation. This suggests that there are significant nonlinear linkages which are not well captured by linear approaches such as WCA.

The application of MSES to the homogeneous regions, obtained using complex network approach, allows unraveling the spatial diversity in the teleconnection patterns over India. ENSO has a strong influence on precipitation in the southeastern parts of the country. These regions are also affected by IOD, however, the IOD influence is much weaker compared to ENSO. NAO has a strong connection to extreme precipitation particularly in the northern regions. The effect of PDO stretches across the whole country, whereas AMO influences precipitation

particularly in the arid and semi-arid regions. The substantial variation of precipitation teleconnections across India and across time scales that is unraveled by the proposed method provides an exciting perspective for rainfall forecasting for India and for making better sense of its weather.

**Data availability.**

The authors used high-resolution (0.25° × 0.25°) monthly gridded precipitation data set developed by the Indian Meteorological Department (IMD) for the spatial domain of 66.5°E to 100°E and 6.5°N to 38.5°N, covering the mainland region of India (Pai et al., 2014). The gridded data has been generated from the observations of 6995 gauging stations across India (Pai et al., 2014). Details about these data can be obtained from http://imd.gov.in
(homepage/Rainfall information).

**Author contribution**
AA devised the project, the main conceptual ideas, and proof outline. AA & RM jointly developed the theoretical framework. AA took lead and implemented the method on real dataset and performs the analysis and tests. AA
arranged, preprocessed the dataset, prepared all the figures and wrote the main test. BM closely supervised the work, helped extensively in rewriting the text for journal submission and helped to improve the figures. Intense discussion with NM & JK helped to reach to appropriate conclusion relevant for the climatological interpretation. RK provided the additional help to draw a meaningful conclusion in term of climatological interpretations. RS and LC helped me in rewriting text, climatological interpretations and conceptualizing figures.

**Competing interests**

The authors declare that they have no conflict of interest.

**Acknowledgements**

This research was funded by the Deutsche Forschungsgemeinschaft (DFG) (GRK 2043/1) within the graduate
research training group "Natural risk in a changing world (NatRiskChange)" at the University of Potsdam (http://www.uni-potsdam.de/natriskchange). The authors gratefully thank the Roopam Shukla and Dr. Bedartha Goswami for the helpful suggestion.

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
