# Peer review of "Unraveling the spatial diversity of Indian precipitation teleconnections via nonlinear multi-scale approach"

_Nonlinear Processes in Geophysics, 2019_

## Referee Comment (RC1) · Anonymous Referee #1 · 29 May 2019

The authors use two approaches namely the Wavelet Coherence Analysis (WCA) and Multi-Scale Event Synchronization (MSES) method to look at Indian precipitation teleconnections. They conclude that MSES is superior to WCA. I have a little problem with this and I would like the authors to address my following observation. If one carefully compares Figures 3 and 4 one will find out that the significant peaks in WCA are almost ALWAYS at lower time scales than the significant time scales in MSES. Why is that? Are we looking at two methods each being more appropriate for a certain time scale range? In this case, what is the point in comparing the two methods?

The above being my major comment, I also have some more comments:

[Figure]

1) In Figure one (upper left) should Even be Event? 2) The 95% confidence levels appear to be similar for all indices in each method. Why should this be so? 3) Figure 2: The community structure is based on Agarwal et al (2018b). How robust is it? The whole paper is based on this community structure. The authors should compare their approach to other approaches in the literature.
* * *

---

## Referee Comment (RC2) · Anonymous Referee #2 · 10 Jun 2019

This paper introduces a nonlinear, multiscale approach, based on wavelets and event synchronization, for unraveling teleconnection influences on precipitation. The results suggest significant nonlinear influences which are not well captured by the wavelet coherence analysis, the state-of-the-art method in understanding linkages at multiple time scales. The results provide an exciting perspective for capturing the dynamics of precipitation and improving precipitation forecasting. In addition, the substantial variation of precipitation teleconnections across India and across time scales that is unraveled by the proposed method provides an exciting perspective for rainfall forecasting for India and for making better sense of its weather. The analysis is very interesting and the results are insightful. The paper is well written and I would like to recommend the

publication of this paper once the following minor points can be addressed:

1) The Z-P space approach should be given some basic descriptions in this paper to help readers understand this paper conveniently. 2) In the Event synchronization and network construction part, 95% threshold is chosen. I think a simple description should be given to the choice criterion.

---

## Author Comment (AC1) · 13 Jul 2019

**Interactive comment on**

**"Unraveling the spatial diversity of Indian precipitation teleconnections via nonlinear multi-scale approach"**

Kurths et al.

Correspondence to: Ankit Agarwal (aagarwal@uni-potsdam.de)

**COMMENTS FROM REVIEWERS**

..............................................................................................................................................................................

Reply to the Review Comments

We thank the editor and reviewer for investing his/her valuable time in our manuscript. We have revised our manuscript, taking into consideration all the review comments. Here, we respond to the specific review comments. In what follows, line numbers correspond to those in the clean version.

We have responded (in black) to each reviewer comment (in red).

**Anonymous Referee #1**

The authors use two approaches namely the Wavelet Coherence Analysis (WCA) and Multi-Scale Event Synchronization (MSES) method to look at Indian precipitation teleconnections. They conclude that MSES is superior to WCA. I have a little problem with this and I would like the authors to address my following observation.

If one carefully compares Figures 3 and 4 one will find out that the significant peaks in WCA are almost ALWAYS at lower time scales than the significant time scales in MSES. Why is that? Are we looking at two methods each being more appropriate for a certain time scale range? In this case, what is the point in comparing the two methods?

We thank the reviewer for the constructive summary of our manuscript and also for his/her critical and supportive suggestions. Your feedback is vitally important to increase the readability of the work.

We would like to bring to the attention of the reviewer that both the methods are trying to capture the connection of precipitation with the climate indices at all the timescales and not restricted to certain timescale range. Indeed, it seems that many of significant peaks in WCA (Fig.3) are at lower timescales however it is observed that many of these peaks are well within the confidence interval indicating that these values are not statistically significant (for instance, C2 peak at 1-2 years scale in Fig.3a etc). Wherever these peaks in Global Wavelet Coherence (Fig.3) are high enough i.e. beyond significant level, the same peak also has been captured in the corresponding MSES measure (for instance, connections in Fig. 3&4 (a) at timescale of 1-2 years).

Further, WCA also captures significant linkages at longer timescales (see Fig. 3a, c, d, and e) but again these peaks are well within the confidence interval indicating that these values are not statistically significant. On the other hand, it is noted that the similar peaks were observed in Figure 4 but the values are statistically significant. This might be attributed to the fact that the WC is linear measure and missed all such higher scales time dependent linkages whereas MSES is a nonlinear measure based on dynamic time delay (not fixed) captures such connections.

**Minor issues:**

(1) In Figure one (upper left) should even be Event?
Corrected!!

(2) The 95% confidence levels appear to be similar for all indices in each method. Why should this be so?
The confidence intervals are estimated using the red colour noise signals having similar statistical properties that of the signals. In this case, we have used 1000 samples of the noise signals for estimating the confidence levels. Since the data (both precipitation and the indices) is normalized, the 95% confidence limits appears to be similar.

(3) Figure 2: The community structure is based on Agarwal et al (2018b). How robust is it? The whole paper is based on this community structure. The authors should compare their approach to other approaches in the literature.

The focus of the present study is to unravel the linkages between Indian extreme precipitation and large-scale climatic patterns. In this study, these linkages between climate indices and precipitation are evaluated on a regional scale instead of single grid scale because towards an interpretation of climate dynamics and teleconnection influence or information transfer, such individual grid points are not the entity of interest, as they do not represent distinct climatological processes (Runge et al., 2015).

For such purpose, homogeneous regions identified by Agarwal et al., (2018b) has been used directly in this study. These homogeneous regions identified by Agarwal et al., (2018b) are based on extreme rainfall events rather than mean rainfall which was the case in many standard classification (Rao et al., 1995). Further, several studies have reported superior performance of complex networks in identifying homogeneous regions compared to more traditional methods, such as the hierarchical clustering algorithm or the information theoretic algorithm (Harenberg et al., 2014). In depth analysis on the statistical, physical and climatological characteristics of each identified region has been provided in Agarwal et al., (2018b) which has been published recently in Journal of Hydrology. Hence, author strongly request not to include further details of community detection in the present study however we clearly refer interested readers to previously published study.

---

## Author Comment (AC2) · 13 Jul 2019

**Interactive comment on**

**"Unraveling the spatial diversity of Indian precipitation teleconnections via nonlinear multi-scale approach"**

Kurths et al.

Correspondence to: Ankit Agarwal (aagarwal@uni-potsdam.de)

**COMMENTS FROM REVIEWERS**

.....

**Reply to the Review Comments**

We thank the editor and reviewer for investing his/her valuable time in our manuscript. We have revised our manuscript, taking into consideration all the review comments. Here, we respond to the specific review comments. In what follows, line numbers correspond to those in the clean version. We have responded (in black) to each reviewer comment (in red).

**Anonymous Referee #2**

This paper introduces a nonlinear, multiscale approach, based on wavelets and event synchronization, for unraveling teleconnection influences on precipitation. The results suggest significant nonlinear influences which are not well captured by the wavelet coherence analysis, the state-of-the-art method in understanding linkages at multiple time scales. The results provide an exciting perspective for capturing the dynamics of precipitation and improving precipitation forecasting. In addition, the substantial variation of precipitation teleconnections across India and across time scales that is unraveled by the proposed method provides an exciting perspective for rainfall forecasting for India and for making better sense of its weather. The analysis is very interesting and the results are insightful. The paper is well written and I would like to recommend the publication of this paper once the following minor points can be addressed.

We thank the reviewer for acknowledging the potential of the method in capturing dynamics of precipitation and his/her critical and supportive suggestions.

The Z-P space approach should be given some basic descriptions in this paper to help readers understand this paper conveniently.

We thank the reviewer for demanding a necessary explanation on Z-P space approach that definitely would increase the understanding and readability of the paper. In the revised version we have included more discussion which read as follows.

In the Event synchronization and network construction part, 95% threshold is chosen. I think a simple description should be given to the choice criterion.

We thank the reviewer for highlighting and demanding an explanation for such an important step. In the revised version we have modified the statement which reads as follows:

A number of criteria have been proposed to generate an adjacency matrix from a similarity matrix, such as a fixed amount of link density (Agarwal, 2019; Agarwal et al., 2018; Stolbova et al., 2014) or fixed thresholds (Donges et al., 2009). Here, we consider a 5% link density since it is a well-accepted criteria in general for the network construction. Also, 95th percentile is a good trade-off between sufficient number of connections and capturing high synchronized connections.

**References**

Agarwal, A.: Unraveling spatio-temporal climatic patterns via multi-scale complex networks, Universität Potsdam., 2019.

Agarwal, A., Maheswaran, R., Marwan, N., Caesar, L. and Kurths, J.: Wavelet-based multiscale similarity measure for complex networks, Eur. Phys. J. B, 91(11), doi:10.1140/epjb/e2018-90460-6, 2018.

Donges, J. F., Zou, Y., Marwan, N. and Kurths, J.: Complex networks in climate dynamics: Comparing linear and nonlinear network construction methods, Eur. Phys. J. Spec. Top., 174(1), 157–179, doi:10.1140/epjst/e2009-01098-2, 2009.

Runge, J., Petoukhov, V., Donges, J. F., Hlinka, J., Jajcay, N., Vejmelka, M., Hartman, D., Marwan, N., Palu?, M. and Kurths, J.: Identifying causal gateways and mediators in complex spatio-temporal systems, Nat. Commun., 6, 8502, doi:10.1038/ncomms9502, 2015.

Stolbova, V., Martin, P., Bookhagen, B., Marwan, N. and Kurths, J.: Topology and seasonal evolution of the network of extreme precipitation over the Indian subcontinent and Sri Lanka, Nonlinear Process. Geophys., 21(4), 901–917, doi:10.5194/npg-21-901-2014, 2014.